# Attitudes of Local Communities towards Marula Tree (*Sclerocarya birrea* subsp. *caffra*) Conservation at the Villages of Ha-Mashau and Ha-Mashamba in Limpopo Province, South Africa

**Ndidzulafhi Innocent Sinthumule *** and **Mbuelo Laura Mashau**

Department of Geography, Environmental Management and Energy Studies, University of Johannesburg,
P.O. Box 524, Auckland Park 2006, South Africa; 201600060@student.uj.ac.za
* Correspondence: isinthumule@uj.ac.za; Tel.: +27-11-559-3810

**Abstract:** The marula tree (*Sclerocarya birrea* subsp. *caffra*), a common species in sub-Saharan Africa, grows naturally in both protected and communal land. Although considerable research has been undertaken on these trees in southern Africa, to the authors' knowledge, the attitudes of local communities towards the protection of marula trees, particularly in communal land, has not been researched. This study intends to fill this gap in knowledge by examining the attitudes of local people towards conservation of marula trees. Studying the attitudes of people can provide insights on how they behave and how they are willing to coexist with *S. birrea*. The case study is set in Limpopo Province of South Africa in the villages of Ha-Mashau (Thondoni) and Ha-Mashamba where marula trees grow naturally. To fulfil the aim of this study, door-to-door surveys were carried out in 2018 and questionnaire interviews were used as the main data collection tool in 150 randomly selected households. The study revealed that local communities in the study area had positive attitudes towards conservation of marula trees. Strategies that are used by local communities to protect marula trees in communal land are discussed.

**Keywords:** attitudes; conservation; communities; marula tree; villages

## 1. Introduction

The focal taxon in this paper, marula tree (*Sclerocarya birrea* subsp. *caffra;* family Anacardiaceae) is a widespread and common species in southern Africa, particularly Zimbabwe, Botswana, Namibia, South Africa, and Swaziland [1]. In South Africa, *S. birrea* is common in the savanna areas of northern province, North West, Mpumalanga and northern KwaZulu–Natal [1,2]. *S. birrea* is a deciduous and fast-growing dioecious species reaching 7–17 m in height [2]. Female *S. birrea* trees generally flower from age 7–10 years but the yield increases as the tree matures [3]. The flowering period is between September and December, with fruits borne yearly from January to March [3,4]. Although the fruit size is variable, they are roughly golf-ball- or plum-sized. Ripe fruits have a thick yellow peel and a translucent whitish flesh which is fibrous and juicy [2]. Archaeological evidence suggests that the fruits of *S. birrea* have been a source of human nutrition since 10,000 years BC, making it one of Africa's botanical treasures [5]. *S. birrea* occurs on a wide range of soil types, although they are most commonly found on well-drained soil in regions with an average annual rainfall of between 200 and 1500 mm [6]. A key factor limiting the distribution of *S. birrea* appears to be its sensitivity to frost [2]. The tree has a spreading crown and the leaves are 8–38 cm long and elliptical in shape with smooth margins [3].

The cultural and socio-economic significance of *S. birrea* is well-documented [2,3,5,7–9]. The literature demonstrates that every part of *S. birrea* including the leaves, stem, roots, branches, and

fruits has huge social, cultural, and economic importance [3,10]. For instance, the fruits are used to make jams and fruit juice [5,11], but most commonly are eaten raw or fermented to make wine or beer popular in rural areas [12,13]. The leaves are eaten by livestock in communal areas and by wildlife in protected areas [14–16]. People also harvest the leaves and the bark for traditional medicines [13,17]. In addition, the kernels are either roasted or eaten raw as a snack by people, and the oils extracted from the kernels are used for a variety of purposes including cooking and cosmetics. The kernels of *S. birrea* have high protein (28–31%) and minerals, especially magnesium, phosphorus, iron, zinc, and copper [3]. The wood is used for fencing, fuel, and the manufacture of carvings [2,5].

The literature describes a number of pieces of research on *S. birrea* conducted both in protected and unprotected areas (communal land); some of these references date as far back as 1906. In protected areas, research on *S. birrea* has predominantly examined the impacts of fire [18,19] and elephants on marula trees [14,16,20]. In communal areas, scholars have primarily studied the traditional uses and commercialization of marula trees [3,5,7,21], as well as their ecology and biology [1,2] and domestication [22,23]. Despite the wealth of research on *S. birrea* in sub-Saharan Africa, the attitudes of local communities towards marula trees in communal land have not been studied. This study intends to fill this gap in knowledge by assessing the attitudes of local communities towards conservation of *S. birrea*. Studies that look at the attitude of people are important for a number of reasons. Firstly, people's attitudes are a major factor in the success of a conservation project or survival of a species [24,25]. Secondly, studies on attitudes of local people can play an important role in understanding their needs and aspirations and in identifying their ideas, suggestions, and opinions regarding protection of species or conservation of natural resources [26,27]. Thirdly, attitudinal studies can disclose the prospects of improving relationships and outreach programmes with local communities [28]. Lastly, understanding local communities' attitudes can provide awareness on how they behave and how they are willing to coexist with a particular species [25]. This may influence policies that can encourage more effective management and planning [27,29].

As documented by many scholars, the assessment of people's attitudes has become an important aspect in many studies dealing with wildlife conservation and natural resource management [25,28,30,31]. The concept of 'attitude' has been used in relation to the positive or negative responses by local people towards one or more stimuli, but can also be linked to possible conduct and behavior [26]. The aim of this study is to explore the attitudes of local communities towards conservation of *S. birrea*. This will help to understand the relationship between local communities and marula trees. The idea is to disclose whether positive or negative attitudes exist towards *S. birrea* that may quantify the role of local communities in conserving the species. This paper is organised into five sections. Section 2 describes the study area and the methods employed in collecting and analysing data. The results are presented in Section 3, while Section 4 presents discussion of the results. Section 5 sets out conclusions of the study.

## 2. Materials and Methods

### 2.1. Study Area

The study area comprises Ha-Mashau (23°8′46.36″ S, 30°11′52.53″ E) and Ha-Mashamba villages (23°14′16.36″ S; 30°8′19.30″ E) situated 20 km apart. The two villages are on communal land falling under the Mashau Tribal Authority. Just like other communal villages in South Africa, the land is owned by the state but is administered by the tribal authority. Ha-Mashau and Ha-Mashamba villages fall in the Makhado Local Municipality within the Vhembe District Municipality in Limpopo Province of South Africa (Figure 1). Although *S. birrea* is a widespread species in southern Africa, it is not found in each and every village. Where it is found, it is not possible to study all the villages. It is for this reason that the villages of Ha-Mashau and Ha-Mashamba where *S. birrea* grow naturally were selected for this study. These two chosen villages are representatives of other communal areas that have marula trees.

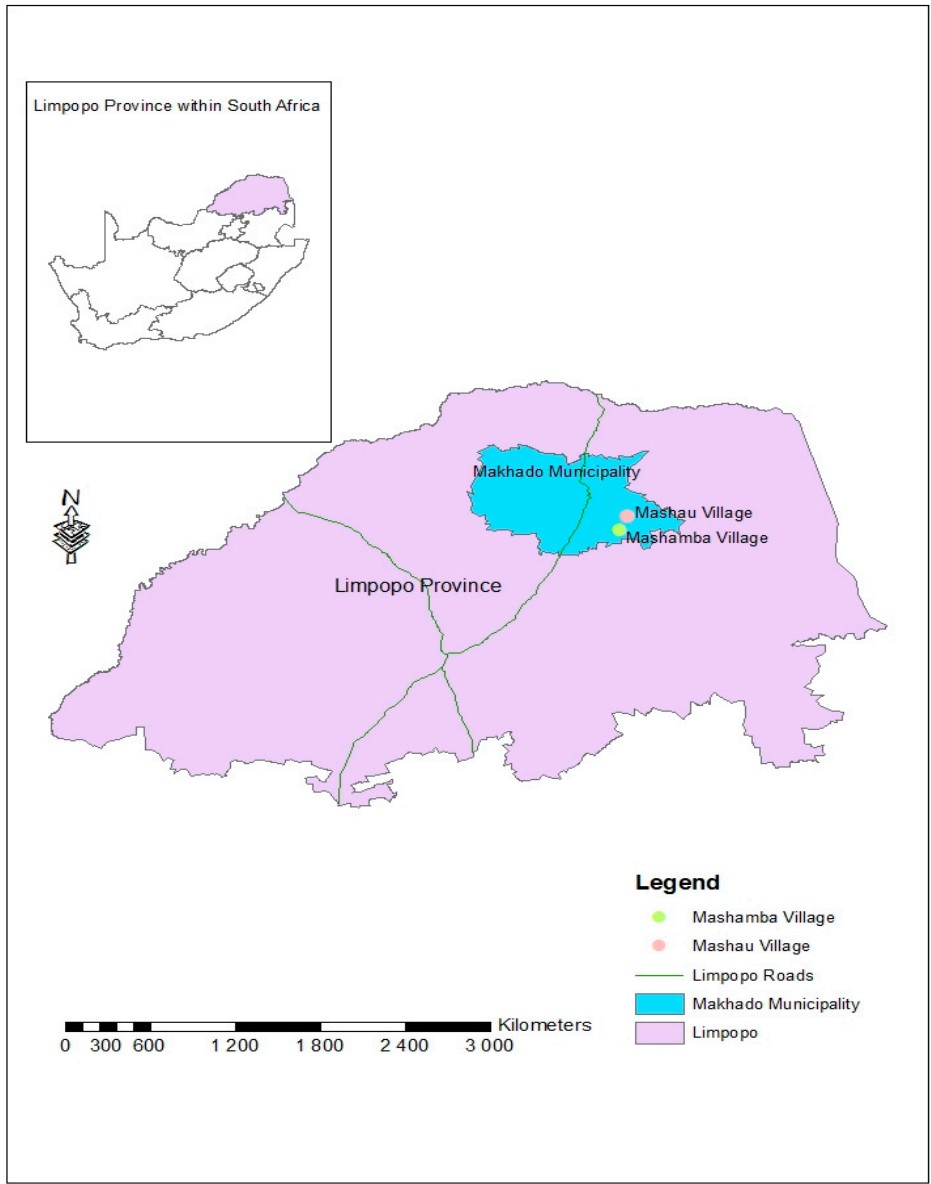

**Figure 1.** Location of the study area.

The study area is located at an altitude of 600–1000 m above sea level [32]. The villages of Ha-Mashau and Ha-Mashamba cover areas of 3.38 km$^2$ and 7.00 km$^2$ of communal land, respectively [33]. The main land uses in the study area include livestock farming, wood harvesting, human settlement, and commercial subsistence agriculture [34]. Local communities in the two villages depend on their communal land for collection of wood for making fire, fencing, building, livestock grazing, and collection of wild fruits.

The two villages fall within a warm temperate, summer rainfall region characterised by cool, frost-free winters and hot summers. The study area receives a mean annual rainfall (MAR) of 550 mm [32]. The dry period is from May to August whereas the rainy season spans the period from December to March. The maximum summer temperature is experienced from September to March with an average high of 30 to 40 °C. The lowest temperatures are experienced between May and August [34].

Ha-Mashau has a population of 2206 (651.83 per km$^2$) distributed across 616 households (182.02 per km$^2$) whereas Ha-Mashamba village has a larger population of 6348 (907.12 per km$^2$)

distributed across 1681 households (240.17 per km$^2$). Ha-Mashau has 1202 (54.49%) females and 1004 (45.51%) males whereas Ha-Mashamba has 3509 (55.27%) females and 2839 (44.72%) males [33]. Although the two villages have electricity, the majority of people still depend on firewood as a principal source of energy. The study area is located in the Tzaneen Sour Bushveld unit (SVI 8) in the Savanna Biome that is the largest in southern Africa. The vegetation in the Tzaneen Sour Bushveld is deciduous, tall open bushveld with a well-developed tall grass layer. The Tzaneen Sour Bushveld is under threat from transformation of the landscape. The dominant tall tree species include *Pterocarpus angolensis,* and *S. birrea* subsp. *caffra* [32].

*2.2. Methods*

The study relied on primary data collected in the study area in September 2018 after gaining permission from the relevant tribal authority. The data were collected through questionnaire interviews, supplemented by observations. Respondents were assured that their answers would be kept confidential; to implement this, the names of interviewees are not revealed in this paper. Furthermore, the research and questionnaire had been vetted and cleared by the tribal authority. A questionnaire was considered to be more appropriate than verbal interviews in this study, because it provides quantifiable answers for a research topic. In addition, it allows data to be collected through both open- and closed-ended questions with all respondents being asked the same questions, thus avoiding interviewer bias [35,36]. The open-ended questions allowed interviewees to express themselves in their own words regarding the knowledge, importance, and attitudes of local people towards conservation of *S. birrea*. Each household was considered a sampling unit, and one questionnaire was restricted to a single respondent per household (preferably the household head; either male or female). Where the household head was absent, any other member 21 years of age or above was considered.

A probability random sampling method that guarantees each household to have an equal chance of selection [37] was used in this study until a maximum of 150 households (75 in each village) had been interviewed. The questionnaires were translated to Tshivenda (the local language of the area) except where respondents were fluent in English. The total response time ranged from 15 to 30 min. Questionnaires were pre-tested [37] on 15 people in a nearby village that was not part of the selected sample prior to the study, which assisted with correcting any ambiguities in the questions [37]. The survey included questions about socio-economic and demographic characteristics, knowledge and importance of marula trees, and attitudes towards their conservation. The assurance that answers would be kept confidential allowed respondents to give their honest opinions. The collected data were analysed using Statistical Package for Social Sciences (SPSS) version 20 (IBM SPSS Inc., Chicago, IL, USA) at a 5% significance level. Descriptive statistics were used to provide a summary of the questionnaire response data set. In order to determine if responses occurred with equal probability, a Chi-square ($\chi^2$) test was applied using Microsoft Excel.

## 3. Results

*3.1. Socio-Economic Profile of the Respondents*

Of the 150 respondents in this study, 84.7% ($n$ = 127) were woman, whereas 15.3% ($n$ = 23) were men. Women and the elderly were generally found to be at home during the survey period and therefore were the respondents for most households. The number of people per household ranged from 1 to 14 (mean = 5.78; SD = 2.43). Of the respondents participated in the survey, 4.7% ($n$ = 7) were 21–30 years old, 14% ($n$ = 21) were 31–40 years of age, 17.3% ($n$ = 26) were 41–50 years of age, 16.7% ($n$ = 25) were 51–60 years, and 47.3% ($n$ = 71) were >60 years ($\chi^2$ = 77.73; df = 4; $P$ < 0.0001). In terms of education, 18.7% ($n$ = 28) had never attended school, 18.7% ($n$ = 28) had attended as far as primary school, 42.7% ($n$ = 64) had secondary school as their highest level of education, and about half that number (20%; $n$ = 30) had tertiary education ($\chi^2$ = 25.04; df = 3; $P$ < 0.0001). With regard to employment, 57.7% ($n$ = 85) were unemployed, 27.3% ($n$ = 41) were employed, and the remaining 14%

($n$ = 24) were self-employed ($\chi^2$ = 39.64; df = 2; $P$ < 0.0001). The study also found that 2.7% ($n$ = 4) had no income at all, 24.7% ($n$ = 37) had an income of R500–R1000, 16.7% ($n$ = 25) had income of R2000–R6000, 6.7% ($n$ = 10) had income of >R6000, whereas the remaining 49.7% ($n$=74) did not specify their income ($\chi^2$ = 102.87; df = 4; $P$ < 0.0001).

Sources of income were found to range from formal employment to home-based micro enterprises such as sewing, upholstery, and welding and state welfare grants (child grant and old age pension). A total of 31.3% ($n$ = 47) of respondents have lived in the area since birth, 7.3% ($n$ = 11) have stayed for 1–10 years, 9.3% ($n$ = 14) have stayed for 21–30 years, 8.7% ($n$ = 13) have stayed for more than 31 years, whereas 43.3% ($n$ = 65) were not sure how long they have stayed in the area ($\chi^2$ = 80.67; df = 4; $P$ < 0.0001).

### 3.2. Knowledge and Importance of Marula Trees

All respondents knew marula trees in the study area but most importantly, they knew that they are indigenous to South Africa. Seventy-three percent ($n$ = 109) were aware that the flowering and fruiting season for marula trees is in summer, whereas the remaining 27% ($n$ = 41) were unsure ($\chi^2$ = 30.83; df = 1; $P$ < 0.0001). Thirteen percent of the respondents ($n$ = 20) had marula trees on their property while the remainder ($n$ = 130) did not but have seen them growing on village communal land. Those who had marula trees in their properties indicated that they did not plant them, and that they were growing naturally. Almost half (46%) of the people questioned stated that they enjoyed drinking marula-derived beer, 13% noted they used it as a source of medicine, 12% enjoyed the marula nuts, 11% enjoyed eating the raw fruits, 8% were found to enjoy the jam while the remaining 4%, 3%, and 2% noted the use of marula as a source of shade, skin lotion, and wood, respectively. Although marula trees are important in the lives of local communities, the majority of respondents (93%; $n$ = 140) indicated that it does not have an economic impact on the lives and livelihoods of local people ($\chi^2$ = 40.56; df = 1; $P$ < 0.0001), and that marula products are not commercialized in the study area.

### 3.3. Attitudes Towards Marula Trees

Almost all respondents (99%; $n$ = 149) were pleased that their village is located within the area dominated by marula trees. A high proportion of interviewees (97%; $n$ = 141) agreed that the marula trees existed for the betterment of local people ($\chi^2$ = 248.92; df = 2; $P$ < 0.0001). When asked if marula trees were the main source of wood in the area, 10% said they were a primary source. Those who used marula trees as a source of wood indicated that they targeted either the male trees that did not bear fruits or dry (dead) marula trees. The majority of respondents (90%; $n$ = 135) stated that neither they harvested marula wood, nor they cut the branches when harvesting fruits to ensure that the trees remained in good condition. Eighty-four percent ($n$ = 126) were of the view that marula trees should be protected wherever they were found. Seventy-eight percent ($n$ = 117) felt, furthermore, that it was their duty to protect marula trees in their villages ($\chi^2$ = 47.04; df = 1; $P$ < 0.0001). Thirty-eight percent ($n$ = 57) indicated that it is important that local people should start planting marula trees on their properties ($\chi^2$ = 77.16; df = 2; $P$ < 0.0001). This is in acknowledgement of the value that is attached to the marula trees. Respondents, particularly those who were above the age of 41 (76%; $n$ = 114), were of the view that fruit trees like marula are traditionally protected because they are a source of food ($\chi^2$ = 123.52; df = 2; $P$ < 0.0001). When asked if there were traditional laws or taboos in the area that prohibit people from cutting marula trees, 75.3% ($n$ = 113) said yes, 14.7% ($n$ = 22) said no, and the remaining 10% ($n$ = 15) were not sure ($\chi^2$ = 119.56; df =2; $P$ < 0.0001). Overall, the majority of the respondents are positive about the protection of marula trees ($n$ = 147; 98%) (Table 1).

**Table 1.** Attitudes of local people towards conservation of marula trees in the study area.

| Attitude Question | Response% | | |
|---|---|---|---|
| | No | Neutral | Yes |
| Do you agree/disagree that the marula trees exists for the betterment of your community? | 5.3 | 0.7 | 97.0 |
| Are you satisfied that your village is located within an area dominated by marula trees? | 0.7 | 00 | 99.0 |
| Do marula trees have an economic impact on the lives and livelihoods of your community? | 76.0 | 00 | 24.0 |
| Do you agree/disagree that marula trees are the main source of wood in your household? | 90.0 | 00 | 10.0 |
| I do not cut the branches of marula trees when harvesting the fruits. | 10.0 | 00 | 90.0 |
| Do you agree/disagree that marula trees must be protected wherever they are found? | 16.0 | 00 | 84.0 |
| Do you agree/disagree that it is a good thing to plant marula trees in your yard? | 60.0 | 00 | 38.0 |
| Fruit trees like marula are traditionally protected because they are a source of food. | 14.7 | 9.3 | 76.0 |
| There are traditional laws or taboo in this village that prohibit people from cutting marula trees? | 14.7 | 10.0 | 75.3 |
| Overall, do you support the conservation of marula trees? | 2.0 | 00 | 98.0 |

## 4. Discussion

This study showsthat the respondents in the villages of Ha-Mashau and Ha-Mashamba have a good knowledge of marula trees and they are aware that the species is indigenous to South Africa. This supports previous studies that explained that *S. birrea* is a tree well-known to local people in southern Africa [2]. Some studies [2,3,5,7,8] documented the importance of this species in the lives of people; hence, *S. birrea* is considered a keystone species [9,21]. For instance, local communities enjoy eating its fruit and nuts, but also use it to a certain extent for fuelwood and medicinal purposes. It is also processed into marula beer, jam, and skin lotion. Furthermore, it provides shade in the heat of the day. Other scholars have similarly found that marula trees play a significant role in the lives and livelihoods of local communities [3,5,8]. However, unlike in other areas such as Swaziland and Bushbuckridge in South Africa [3,17,21], marula products have not been commercialised in Ha-Mashau and Ha-Mashamba villages. In the present study, the majority of respondents indicated that marula trees do not have an economic impact on the lives and livelihoods of people, because marula products (e.g., marula beer) are enjoyed for free when there are social gatherings. Indeed, marula beer serves to strengthen social bonds with neighbors, friends, and relatives and helps to build a united community [5]. Similarly, its shade fosters social gathering and a sense of place.

The majority of the people are happy to live in an area dominated by marula trees, which they see as a blessing because of its importance in their lives. As a result, most respondents do not use marula trees as a source of wood despite the fact that fuelwood remains the principal source of energy in the area. This result is consistent with other studies that reported felling of marula trees in most rural society was strictly a taboo [7]. Those who use marula trees as a source of wood target dead wood and males specimens that do not bear fruits. Similar results were also found in Matiyane Village [38]. Furthermore, most respondents do not cut the branches of trees when harvesting marula fruits. This is in line with other studies that recounted that only fruits that had fallen to the ground and ripened are collected, as harvesting directly from the marula tree is perceived as being a taboo [5]. Unlike in protected areas where marula trees are regularly destroyed by elephants [16,20], the villages of Ha-Mashau and Ha-Mashamba were both found to have tall marula trees (particularly female specimens) despite being a common property resource. This finding is contrary to some scholars who equate local people as being agents of environmental destruction [39]. In addition, the significance of marula trees for local people has led to the domestication of the tree, and hence people have tenurial rights over marula trees in their homesteads. However, unlike in other areas such as Matiyane and Bushbuckridge villages in South Africa where local communities have cultivated marula seedlings in their yards [7,38,40], this study found that people do not propagate marula trees. The trees in properties were self-seeded and had grown naturally. On the other hand, the majority of respondents were of the view that local people should indeed start planting marula trees on their properties, attesting the value that people attach to these trees [3,21].

A high proportion of respondents were of the view that marula trees must be protected wherever they were found. Furthermore, most respondents irrespective of education level or gender felt

that it was their responsibility to protect marula trees in their villages. Respondents over 40 years of age indicated that fruit trees like marula were traditionally protected from any harm; as such a taboo prohibits people in the area from felling female marula trees. In other words, there are locally recognised traditional beliefs and more specific norms or customs that are orally held by local communities [41]. There is no written documentation [42] and such traditional beliefs are transferred from older to younger generation. Local community institutions such as the tribal authorities help to enforce the traditional norms or rules [41] for the protection of marula trees. Although most respondents reported that there are traditional laws that disallow the destruction of fruit trees like marula, they also pointed out that traditional authorities no longer enforce those laws. This implies that the law does not punish people for cutting marula trees, but that local residents are alert to the destruction of marula trees in the area. Thus, even though there are no rangers or other security in these villages to guard against the destruction of natural resources, the trees remain intact.

## 5. Conclusions

This study has demonstrated that local communities in the villages of Ha-Mashau and Ha-Mashamba are reliant on *S. birrea* for survival. The reliance on plants has often resulted in decrease of forest and woodland tree cover species [39,43]. Surprisingly, the results of this study suggest that marula trees are well-conserved despite the threats to conservation of the Tzaneen Sour Bushveld as a whole [32]. Contributing to the protection of the species is the value placed on the services of the marula by local people. A high proportion of respondents still believe in the tradition that fruit trees like marula must not be cut, as they are an important source of food, and that the protection of marula trees is their responsibility. These results inform conservationists and policy-makers about the importance of traditional beliefs or customs in conservation and may influence policies that can encourage a more effective management and planning. Senior citizens or pensioners should be encouraged to teach younger generations about indigenous knowledge that promote the protection of marula trees. This study has found that positive attitudes towards marula trees held by the majority of respondents contribute to the successful conservation of the species. It is important to recognise and foster such positive attitudes for sustainable natural resource management, and as shown in this study even communities or individuals can support conservation and contribute towards the protection of natural resources, provided that they get tangible benefits from their surrounding environment.

**Author Contributions:** Conceptualization, N.I.S.; methodology, N.I.S. and M.L.M; software, N.I.S. and M.L.M; validation, N.I.S. and M.L.M; formal analysis, M.L.M and N.I.S.; investigation, M.L.M.; resources, M.L.M and N.I.S.; data curation, N.I.S. and M.L.M.; writing—original draft preparation, N.I.S.; writing—review and editing, N.I.S.; visualization, N.I.S.; supervision, N.I.S; project administration, N.I.S. and M.L.M.

**Funding:** This research received no external funding.

**Acknowledgments:** The authors gratefully acknowledge all the respondents in the villages of Ha-Mashau (Thondoni) and Ha-Mashamba who voluntarily participated in this study. Mbuelo Laura Mashau appreciate the support from her parents; J.V Mashau and T Mashau for encouraging her to work hard regardless of challenges. We are very grateful to the anonymous reviewers whose comments and remarks contributed to the improvement of the quality of the paper.

**Conflicts of Interest:** The authors declare there are no potential conflicts of interest with respect to the research, authorship, and/or publication of this article.

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
