# Peer review of "Attitudes of Local Communities towards Marula Tree (Sclerocarya birrea subsp. caffra) Conservation at the Villages of Ha-Mashau and Ha-Mashamba in Limpopo Province, South Africa"

_resources, doi:10.3390/resources8010022_

Round 1

Reviewer 1 Report

An interesting study to ascertain community views on an important keystone species. I felt that much of the Results and Discussion could be streamlined a little by avoiding repetition of the results. Also, Table 1 could be safely deleted without loss as all of that information is contained in the text discussion (or, cut back most of the text and just refer the reader to Table 1). With some tight editing this will be an improved paper.

Author Response

Reviewer 1

Reviewer comment: The details about the study species appearing in page 4 must be placed at the start of Introduction.

Response: The whole paragraph on the characteristics of Sclerocarya birrea have been copied and pasted in the introduction section. It now start the Introduction section.

Reviewer comment: I felt that much of the Results and Discussion could be streamlined a little by avoiding repetition of the results. Table 1 could be safely deleted without loss as all of that information is contained in the text discussion (or, cut back most of the text and just refer the reader to Table 1).

Response: Table 1 deleted.

Reviewer comment: With some tight editing this will be an improved paper.

Response: The paper has been edited.

Reviewer 2 Report

“Attitudes of local communities towards marula tree (Sclerocarya birrea subsp. caffra) conservation at the villages of Ha-Mashau and Ha-Mashamba in Limpopo Province, South Africa” 

This represents an interesting paper about an important traditional resource in Africa. It has novel information about attitudes of local residents in two villages and it also provides a revision of what has been said about the species in other papers.

I suggest improving the paper in the following aspects:

In the abstract: Briefly include the abstract the reason why studying the attitudes of local communities towards the tree is important (in no more than a sentence).

Include an explanation of the relation of the site to the species (is it a site representative of its widespread range?). Why are does villages relevant for this study? Why where they chose? And also mention if they are representative of other areas. 

A little about the natural history of the species can aid in understanding the relevance of the study site chosen.

Explain under what type of protection is the species in the introduction. Also explain what it is meant by “traditionally protected” 

Comparison between female and male respondents is not possible given the unbalance number of people interviewed. Therefore affirmation such as “…the villages of Ha-Mashau and Ha-Mashamba were both found to have tall marula trees (particularly females) despite being a common property resource..” (or is the gender referring to the tree?). This is not clear

Why are local communities impoverished? This is mentioned in the conclusions but not in the study site.

It would be important for the authors to provide what of the findings from this research can be incorporated into policy. Would it be important to foment the seeding of the species? Provide incentives for the commercialization of products? Or is this not important.

Author Response

Reviewer 2

“Attitudes of local communities towards marula tree (Sclerocarya birrea subsp. caffra) conservation at the villages of Ha-Mashau and Ha-Mashamba in Limpopo Province, South Africa” 

This represents an interesting paper about an important traditional resource in Africa. It has novel information about attitudes of local residents in two villages and it also provides a revision of what has been said about the species in other papers.

I suggest improving the paper in the following aspects:

Reviewer comment: In the abstract: Briefly include the abstract the reason why studying the attitudes of local communities towards the tree is important (in no more than a sentence).

Response: A sentence including the importance of studying the attitudes of local communities has been included in the abstract (page 1).

Reviewer comment: Include an explanation of the relation of the site to the species (is it a site representative of its widespread range?). Why are does villages relevant for this study? Why where they chose? And also mention if they are representative of other areas. 

Response: Information or reasons on why the two villages of Ha-Mashau and Ha-Mashamba were selected/chosen for this study is provided in page 3.  

Reviewer comment: Explain under what type of protection is the species in the introduction. Also explain what it is meant by “traditionally protected”

Response: Description on what is meant by “traditionally protected” is given in page 9 and 10.

Reviewer comment: Comparison between female and male respondents is not possible given the unbalance number of people interviewed. Therefore affirmation such as “…the villages of Ha-Mashau and Ha-Mashamba were both found to have tall marula trees (particularly females) despite being a common property resource..” (or is the gender referring to the tree?). This is not clear

Response: The sentence has been clarified in page 9.

Reviewer comment: Why are local communities impoverished? This is mentioned in the conclusions but not in the study site.

Response: The sentence has now been removed in the conclusion.

Reviewer comment: It would be important for the authors to provide what of the findings from this research can be incorporated into policy. Would it be important to foment the seeding of the species? Provide incentives for the commercialization of products? Or is this not important.

Response: This has now been included in the conclusion which is in page 10.